# Spatiotemporal properties of glutamate input support direction selectivity in the dendrites of retinal starburst amacrine cells

Prerna Srivastava[1†], Geoff de Rosenroll[1†], Akihiro Matsumoto[2], Tracy Michaels[1], Zachary Turple[1], Varsha Jain[1], Santhosh Sethuramanujam[1], Benjamin L Murphy-Baum[1], Keisuke Yonehara[2], Gautam Bhagwan Awatramani[1*]

[1]Department of Biology, University of Victoria, Victoria, Canada; [2]Danish Research Institute of Translational Neuroscience, Nordic-EMBL Partnership for Molecular Medicine, Department of Biomedicine, Aarhus University, Aarhus, Denmark

**Abstract** The asymmetric summation of kinetically distinct glutamate inputs across the dendrites of retinal 'starburst' amacrine cells is one of the several mechanisms that have been proposed to underlie their direction-selective properties, but experimentally verifying input kinetics has been a challenge. Here, we used two-photon glutamate sensor (iGluSnFR) imaging to directly measure the input kinetics across individual starburst dendrites. We found that signals measured from proximal dendrites were relatively sustained compared to those measured from distal dendrites. These differences were observed across a range of stimulus sizes and appeared to be shaped mainly by excitatory rather than inhibitory network interactions. Temporal deconvolution analysis suggests that the steady-state vesicle release rate was ~3 times larger at proximal sites compared to distal sites. Using a connectomics-inspired computational model, we demonstrate that input kinetics play an important role in shaping direction selectivity at low stimulus velocities. Taken together, these results provide direct support for the 'space-time wiring' model for direction selectivity.

*For correspondence:
gautam@uvic.ca

†These authors contributed equally to this work

Competing interest: The authors declare that no competing interests exist.

## Editor's evaluation

This is an important paper that addresses a key mechanism that underlies the canonical computation of direction selectivity in the retina. By using fluorescence imaging of glutamate release from excitatory interneurons combined with a computational model of dendritic integration, the authors make a convincing case that the kinetics of glutamate release contributes to the direction-selectivity of individual neural processes in retinal neurons. This work will appeal to visual neuroscientists as well as cellular physiologists interested in dendritic computations.

## Introduction

The radiating dendrites of retinal GABAergic/cholinergic 'starburst' amacrine cells (starbursts) are the first points in the visual system to exhibit direction selectivity (*Euler et al., 2002*). Object motion away from the soma generates large calcium responses in distal starburst dendrites, from where they release GABA and acetylcholine (ACh). By contrast, motion toward the soma evokes weak responses. Starbursts play a critical role in shaping direction-selective (DS) responses of downstream ganglion cells (DSGCs), and thus, understanding how they compute direction is of principal interest. Decades of intense investigations have identified several mechanisms that underlie direction selectivity in

starburst dendrites, although no single mechanism alone may be critically required (*Ding et al., 2016*; *Hausselt et al., 2007*; *Kim et al., 2014*; *Hanson et al., 2019*; reviewed by *Murphy-Baum et al., 2021*). A model that has garnered recent attention relies on the kinetic properties of distinct sources of glutamatergic input, which is referred to as the 'space-time wiring' model for direction selectivity (*Greene et al., 2016*; *Kim et al., 2014*). In this study, we sought to evaluate the kinetics of glutamatergic input to the ON starburst dendrites to understand their role in generating direction selectivity.

The space-time wiring model for direction selectivity is inspired by connectomic analysis in the mouse retina showing that the proximal and distal dendritic regions of ON and OFF type starburst amacrine cells receive synaptic inputs from anatomically distinct types of glutamatergic bipolar cells (BCs) (*Greene et al., 2016*; *Kim et al., 2014*). As the axon terminals of these BCs stratify at distinct depths within the inner plexiform layer (IPL; *Ding et al., 2016*; *Greene et al., 2016*; *Kim et al., 2014*)—and in general BC axonal stratification patterns are linked to their kinetic properties (*Awatramani and Slaughter, 2000*; *Baden et al., 2016*; *Franke et al., 2017*; *Gaynes et al., 2022*; *Strauss et al., 2022*)—it has been hypothesized that different types of BCs contacting proximal and distal starburst dendrites have distinct response kinetics (*Greene et al., 2016*; *Kim et al., 2014*). Specifically, it is predicted that proximal inputs near the starburst soma are mediated by BC types that support tonic patterns of glutamate release, while distal inputs are mediated by BC types that release their vesicles more transiently. This arrangement would result in an optimal input summation along starburst dendrites during centrifugal (soma-to-dendrite) motion, as experimentally noted.

The space-time wiring model for direction selectivity is algorithmically similar to classic correlation-type motion detectors described elsewhere in the visual system in both rodents and primates (*Lien and Scanziani, 2018*; *De Valois and Cottaris, 1998*), as well as the fly optic lobe (*Haag et al., 2016*; *Leong et al., 2016*; *Behnia et al., 2014*), and thus appears to reflect a core computational principle. While it is an attractive model for direction selectivity, there is scant evidence that the kinetics of glutamatergic input varies along the length of starburst dendrites. Direct electrophysiological measurements revealed that BC5s (including types 5i, o, and t) and BC7, which make 'ribbon' synapses predominantly on the distal and proximal dendrites of ON starbursts, respectively, have similar temporal properties (*Ichinose et al., 2014*). Regional differences in input kinetics were noted when BC output was measured postsynaptically using voltage-clamp techniques (*Fransen and Borghuis, 2017*; but see *Stincic et al., 2016*). However, these electrophysiological recordings do not provide precise information regarding the anatomical location of synaptic inputs.

To this end, genetically encoded fluorescent glutamate sensors (iGluSnFRs) have provided an alternate way to measure BC output kinetics (*Borghuis et al., 2013*; *Marvin et al., 2013*; *Yonehara et al., 2013*). However, recent imaging studies have found that most BCs of the same polarity (including the BC5s and BC7 types) have similar temporal properties, at least to spatially restricted stimuli that are relevant to local starburst dendritic computations (*Franke et al., 2017*; *Strauss et al., 2022*). Finally, optogenetic studies have demonstrated that direction selectivity remains intact under conditions in which all BC input to starbursts is pharmacologically blocked and the starburst network is directly stimulated in relative isolation (*Hanson et al., 2019*; *Sethuramanujam et al., 2016*). Taken together, previous results provide little support for the space-time wiring model for direction selectivity.

In the present study, we identify specific stimulus conditions under which stark kinetic differences can be observed in proximal and distal BC inputs. We did so by directly monitoring input kinetics across starburst dendrites using iGluSnFRs selectively expressed in these cells, across a range of stimuli and pharmacological conditions. We used temporal deconvolution to estimate vesicle release dynamics, and in a connectomics-inspired computational model, we tested how the specific spatial distributions of kinetically distinct glutamatergic inputs impact direction selectivity. We found that diverse BC kinetics play a role in shaping direction selectivity mainly in the context of relatively large objects, which move slowly across starburst's receptive field.

## Results

### Temporal diversity of glutamate responses along single starburst dendrites

We injected AAVs containing flex-iGluSnFR intravitreally into ChAT-Cre-expressing mice. In a few cases, we observed a strong but sparse expression of iGluSnFR in ON-type starburst cells (*Figure 1A*). This

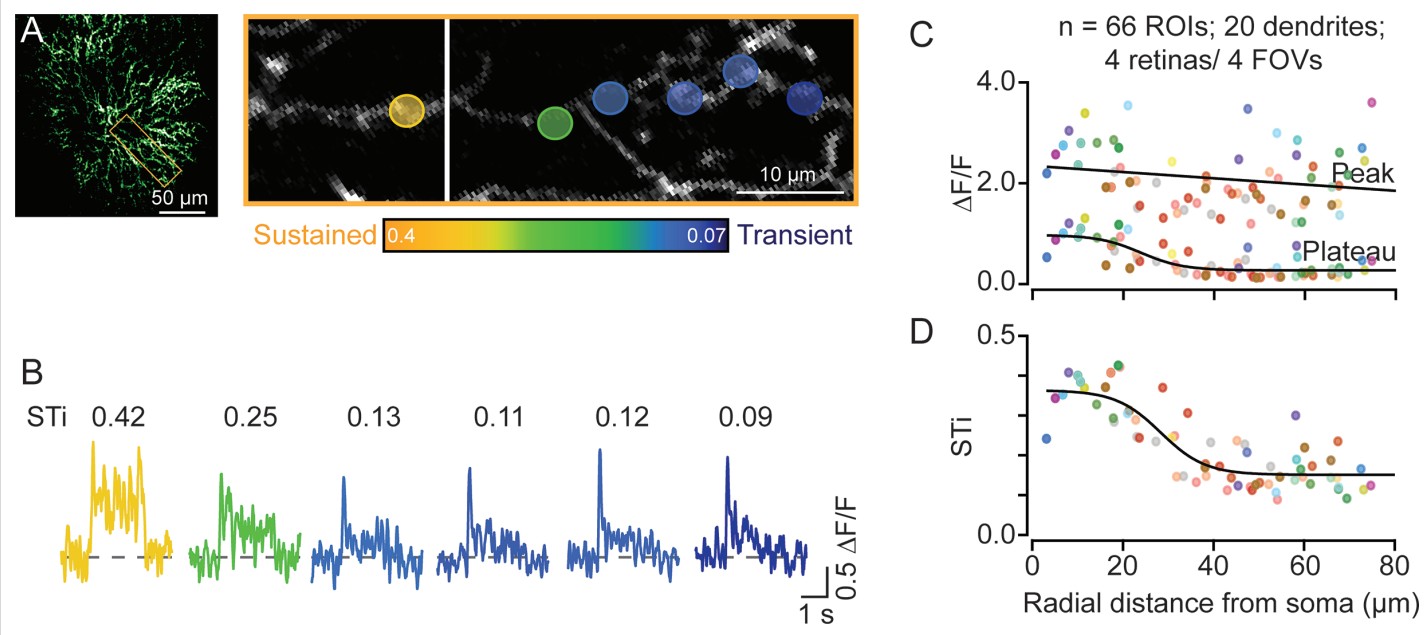

**Figure 1.** Temporal diversity of inputs across single starburst dendrites revealed by sparse iGluSnFR imaging. (**A**) Two-photon z-stack image (left) of a single ON starburst amacrine cell expressing iGluSnFR. Changes in iGluSnFR fluorescence evoked by a 200-μm spot were measured across the single starburst dendrite (yellow box; left). Proximal and distal responses were captured in separate focal planes, and the resulting images were stitched together (right; the vertical white line separates the two planes). (**B**) Examples of time-varying iGluSnFR signals (ΔF/F) (average; two trials) measured in small dendritic regions of interest (5×5 μm² ROIs; shown in (**A**)). The responses and ROIs are color-coded according to their sustained/transient indices (STis; color scale bar shown in (**A**)). The STis (mean; two trials) are indicated above each trace. (**C**) The amplitudes of the peak and plateau iGluSnFR responses are plotted as a function of radial distance from the soma. Each point indicates the value obtained from an individual ROI averaged over two trials; ROIs on the same dendrite share the same color (n=66 ROIs from 20 dendrites/4 retinas/4 mice). (**D**) STis computed from (**C**) plotted as a function of radial distance from the soma. ROI, region of interest.

The online version of this article includes the following source data for figure 1:

**Source data 1.** Temporal diversity of inputs across single starburst dendrites revealed by sparse iGluSnFR imaging.

provided a unique opportunity to visualize glutamate response kinetics across the length of individual dendrites (*Figure 1B*). Signals from dendritic regions proximal to the starburst soma were captured in a different optical plane than the more distal dendrites, which are ~5 μm apart (*Ding et al., 2016*; *Greene et al., 2016*). Spots of light (200 μm in diameter) centered on the imaging field evoked robust iGluSnFR signals throughout the first ~60–80 μm section of starburst dendrites, where glutamatergic BCs are known to make synapses (*Ding et al., 2016*; *Greene et al., 2016*). The peak amplitudes of the iGluSnFR signals measured over small regions of interest (ROIs; 5×5 μm²) were relatively stable across the length of single starburst dendrites (*Figure 1C*). However, the magnitude of the sustained phase significantly decreased as a function of distance (*Figure 1C*; ΔF/F=0.80±0.29 at proximal sites; ΔF/F=0.29±0.16 at distal sites, measured in 20 dendrites in 4 retinas from 4 mice; *p<0.001, t-test). As a result, the sustained/transient index (STi) computed from the plateau/peak ratio, systematically decreased with distance from the soma (*Figure 1D*; STi=0.33±0.06 for proximal, 0.16±0.05 for distal ROIs; *p<0.001, t-test) (Note, STi=0 indicates a purely transient response with no plateau phase, and STi=1 indicates a purely sustained response where peak and plateau phases are equal). Taken together, these results provide the first piece of direct evidence that the kinetics of glutamatergic input varies along starburst dendrites, supporting the 'space-time' wiring model for direction selectivity (*Kim et al., 2014*; *Greene et al., 2016*).

In most experiments, iGluSnFR expression was more widespread across the starburst population. Individual dendrites leaving the starburst soma were easily visible, but as they dove deeper into the IPL, they merged with dendrites from other starbursts to form the intricate 'honeycomb' mesh that is characteristic of these cholinergic cells. By taking care to lay the retina down flat in the recording chamber, we were able to measure responses from proximal and distal starburst dendrites in separate imaging planes (*Figure 2A*). We found that STis were significantly lower in imaging planes

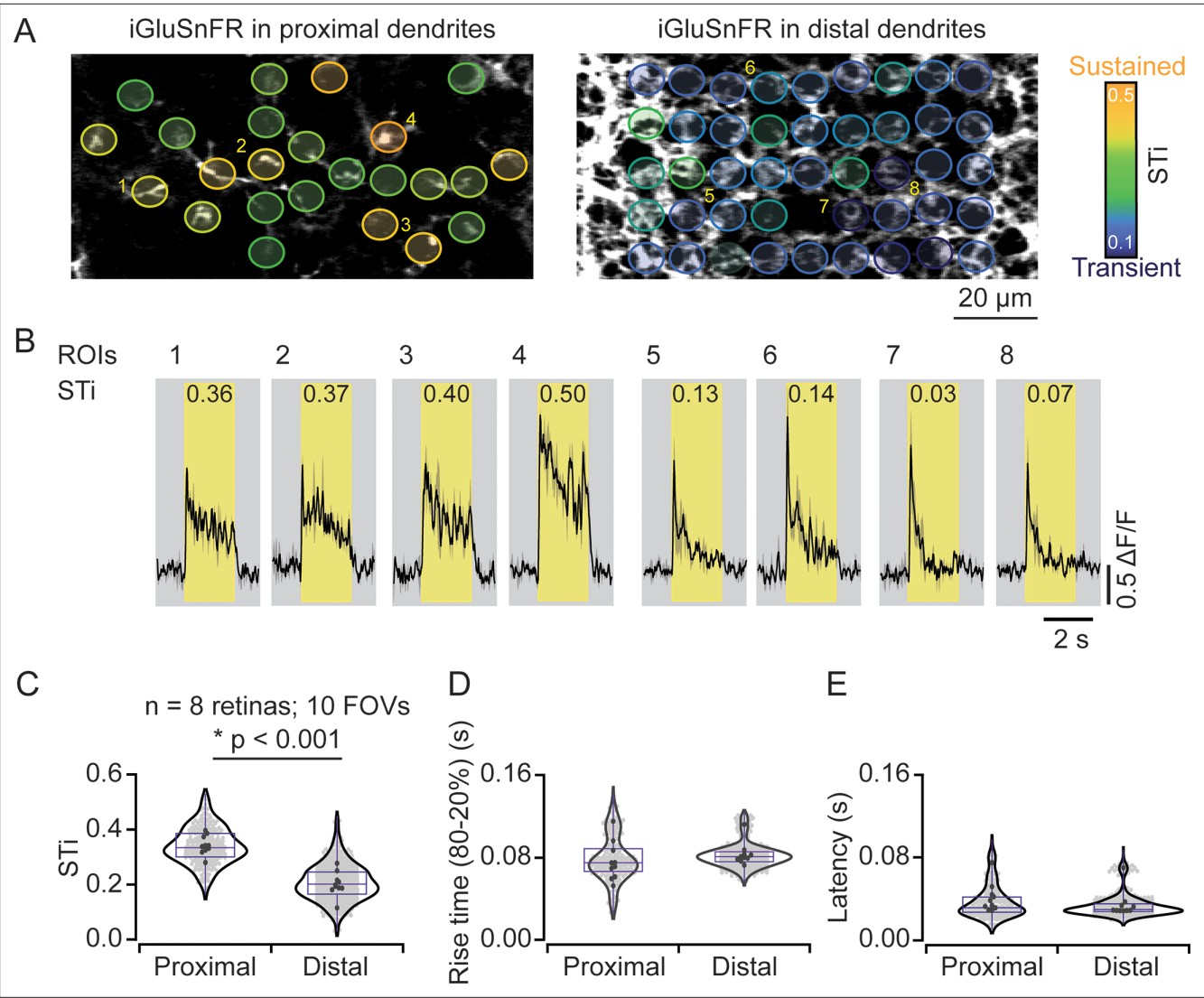

**Figure 2.** Measuring inputs kinetics in the starburst population. (**A**) In the left scan field, proximal dendrites arising from the starburst soma expressing iGluSnFR can be visualized in relative isolation. Images that were taken ~5 μm deeper in the retina (right) reveal the dense 'honeycomb' structure formed by distal starburst dendrites. (**B**) Example iGluSnFR responses evoked by 200 μm static spot extracted for a few ROIs numbered in (**A**) with their STis indicated on the top. Yellow bands indicate stimulus duration. Black, mean responses; gray, ± s.e.m. of two trials. (**C–E**) Distribution of STis (**C**), 80–20% rise times (**D**) and latencies (**E**) in the proximal and distal field of views (FOVs) of the individual (gray) and average (black) ROIs from different recordings (n=10 FOVs, 8 retinas, *p<0.001; t-test). ROI, region of interest; STi, sustained/transient index.

The online version of this article includes the following source data for figure 2:

**Source data 1.** Measuring inputs kinetics in the starburst population.

that captured distal dendrites (STi=0.21±0.07; μ±s.d.); compared to those that captured proximal dendritic responses (STi=0.34±0.07; μ±s.d.) (n=242 proximal ROIs; n=563 distal ROIs; 8 retinas from 8 mice; 10 FOVs; **\***p<0.001, t-test; *Figure 2A–C*), verifying our initial findings on a larger population level. The STi for individual ROIs measured across the population was independent of response amplitude, indicating that the estimated differences in signal kinetics are not strongly compromised by signal-to-noise and/or sensor saturation issues (*Figure 3—figure supplement 1*). In addition, we also found that the latencies and rise times for responses measured at proximal and distal sites were similar (*Figure 2D and E*), indicating that the small differences in axonal path lengths between proximal and distal BCs do not result in significant transmission delays, as previously envisioned (*Kim et al., 2014*).

Anatomical studies show that inputs to proximal starburst dendrites originate mainly from BC7s, while inputs to distal dendrites arise from BC5s (including types BC5i, o, and t; *Greene et al., 2016*;

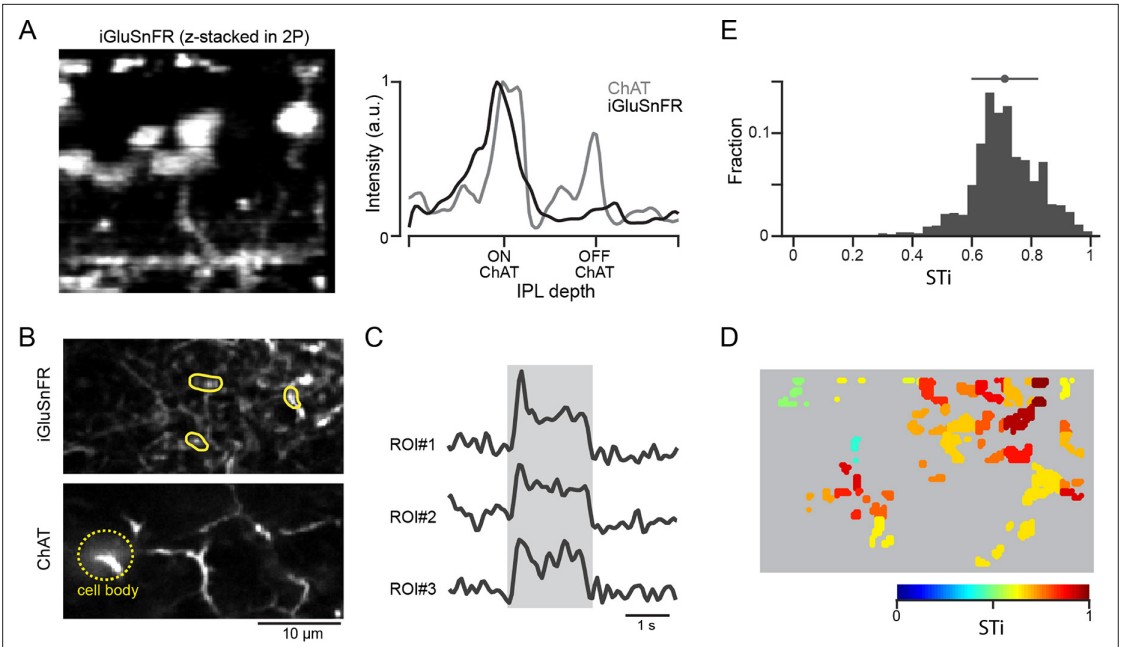

**Figure 3.** Expressing iGluSnFR in BC7 axon terminals reveals their sustained output. (**A**) Cross-section of an image stack showing iGluSnFR labelled BC7 (left). The intensity profiles of the BC terminals labeled with iGluSnFR (gray), and starburst dendrites labeled with tdTomato (black) across the inner plexiform layer (IPL) show that BC terminals co-stratify with dendrites of ON starbursts (right). (**B**) iGluSnFR expression in BC7 terminals (top) imaged at the same depth as the proximal ON starburst dendrites labeled with tdTomato (bottom). (**C**) Light-evoked glutamate signals (right) extracted from three ROIs shown in (**B**) (left). The gray band indicates the stimulus duration. (**D**) Heat maps of the STis for all identified ROIs. (**E**) A histogram of STis for the light-evoked responses for all ROIs. Top, mean (circle), and s.d. (horizontal bar) of the STis. BC, bipolar cell; ROI, region of interest; STi, sustained/transient index.

The online version of this article includes the following source data and figure supplement(s) for figure 3:

**Figure supplement 1.** Response kinetics are not strongly associated with their peak amplitudes.

**Figure supplement 1—source data 1.** Response kinetics are not strongly associated with their peak amplitudes.

**Figure supplement 2.** Temporal properties of proximal and distal input revealed by noise analysis.

**Figure supplement 2—source data 1.** Bipahsic peak and area indices.

*Ding et al., 2016*), indicating that the kinetic differences of iGluSnFR responses may reflect the properties of distinct BC types. By using an AAV-8BP/2 vector containing a CAG promotor, we directly monitored glutamate release at BC7s axon terminals (*Matsumoto et al., 2021*; *Figure 3*). We identified axon terminals of BC7s based on the depth of ON starburst cell dendrites that were genetically labeled by tdTomato (*Matsumoto et al., 2021*). We found that iGluSnFR responses at BC7 terminals were sustained, regardless of their peak amplitude (*Figure 3—figure supplement 1*). The most appreciable changes in iGluSnFR fluorescence occurred at the axon terminals, suggesting that the sensor signals reflect the vesicle release dynamics of individual BC7s (*James et al., 2019*). While these results lend support to the idea that BC7s are the source of sustained proximal input, we were unable to express the sensor in BC5s and could not directly confirm that these were the sources of transient signals observed in distal starburst dendrites.

When white-noise stimuli were used to characterize the temporal properties of BCs using reverse-correlation techniques, we failed to observe significant kinetic differences in proximal and distal iGluSnFR responses. We found the input impulse responses were biphasic for both proximal and distal inputs (*Figure 3—figure supplement 2*). Thus, the probability of glutamate release from BC terminals appears to be transiently depressed, following a burst of vesicle release, during continuous stimulus regimes. Similar biphasic kernels have been observed in recent imaging studies (*Franke et al., 2017*; *Strauss et al., 2022*). As the biphasic nature of the distal—but not proximal—input is critical to the success of models generating direction selectivity (*Kim et al., 2014*; *Fransen and Borghuis, 2017*), we conclude that under conditions where the circuit is continually stimulated at

high frequencies, input kinetics are unlikely to play a role in shaping direction selectivity in starburst dendrites.

## BC output kinetic differences are shaped largely by excitatory network mechanisms

Next, to investigate whether cell-intrinsic or network mechanisms shape BC kinetics, we examined how iGluSnFR responses were affected by stimulus size. Increasing the spot diameter systematically

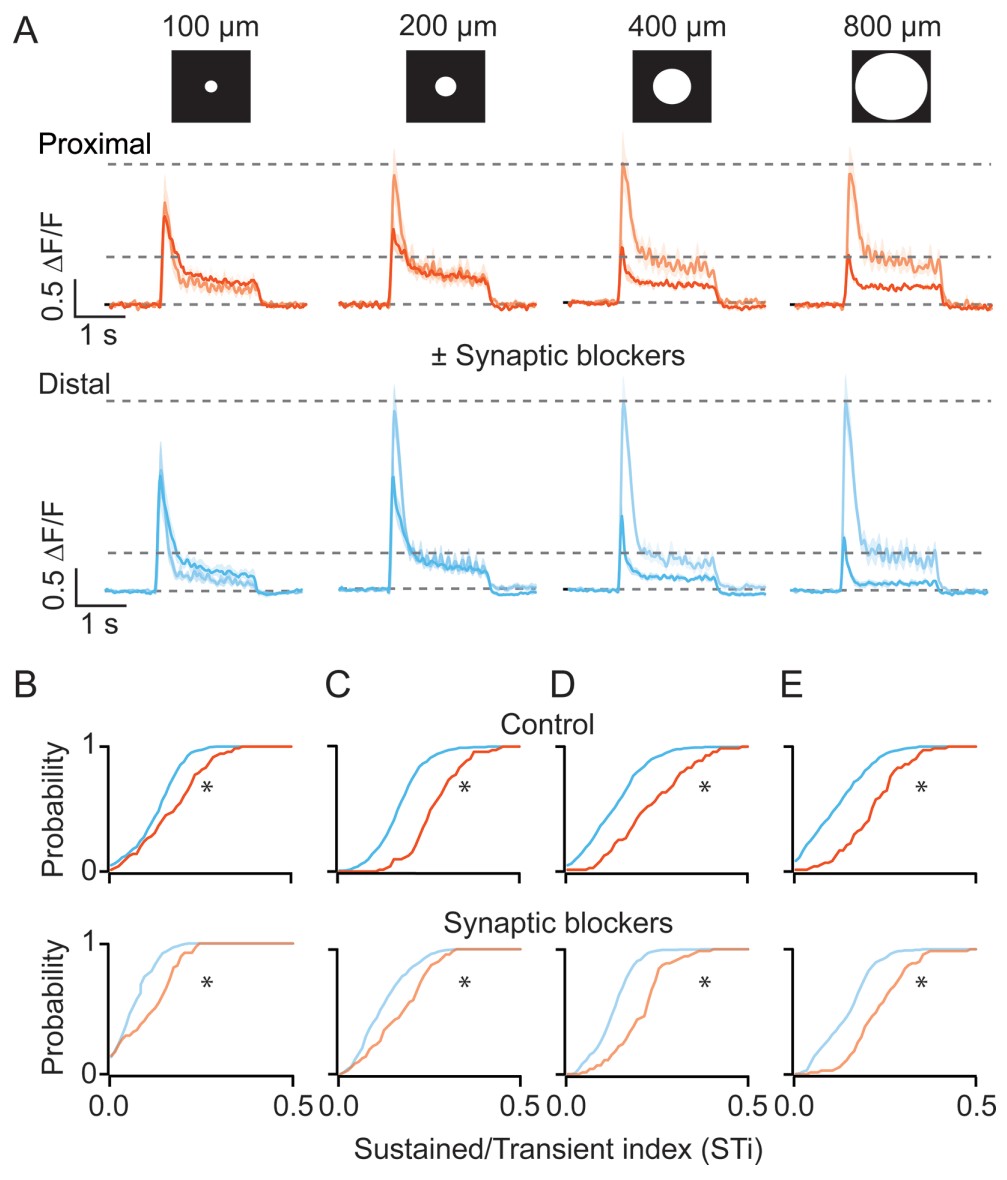

**Figure 4.** Kinetic differences in iGluSnFR signals are apparent across a range of stimulus sizes and persist in the presence of inhibitory receptor blockers. (**A**) The average iGluSnFR signals were evoked by spots of different diameters (100–800 μm). Responses were averaged across five proximal (orange) or distal (blue) FOVs. Responses measured under control (dark traces) conditions and in the presence of synaptic blockers (SR, TPMPA, and CNQX; light traces) are overlaid. Shading indicates ± s.e.m. (**B–E**) Cumulative distributions of STis for ROIs in the proximal and distal FOVs under control and blocker conditions for different stimulus sizes. (n=5 FOVs, 4 retinas, *p<0.001; Kolmogorov-Smirnov test). FOV, field of view; ROI, region of interest; STi, sustained/transient index.

The online version of this article includes the following source data for figure 4:

**Source data 1.** Kinetic differences in iGluSnFR signals are apparent across a range of stimulus sizes and persist in the presence of inhibitory receptor blockers.

decreased the peak amplitude of BC responses, indicative of the recruitment of the inhibitory surround (*Figure 4A*; *Franke et al., 2017*). Importantly, the distinction in the kinetics of proximal and distal inputs remained clear across stimulus sizes, although they were generally more pronounced for stimuli that were >200 µm (*Figure 4B–E*, control; n=71 proximal and 431 distal ROIs, 5 FOVs, 4 retinas from 4 mice; *p<0.001, Kolmogorov-Smirnov test). Indeed, the application of a cocktail of antagonists containing both GABA and ionotropic glutamate receptor antagonists (5 µM gabazine and 100 µM TPMPA, 20 µM CNQX, respectively)—which blocks inhibitory inputs from amacrine and horizontal cells—augmented responses, especially those evoked by larger spots (*Figure 4A*). These effects reduced the overall STi as compared to control, but the kinetics of the iGluSnFR responses at proximal and distal ROIs remained distinct (*Figure 4B–E*, drug cocktail; n=71 proximal and 431 distal ROIs, 5 FOVs, 4 retinas from 4 mice; *p<0.001, Kolmogorov-Smirnov test). Thus, while the inhibitory networks modulate BC responses, they do not appear to account for the sustained/transient differences observed here.

Blocking glutamate/GABA receptor-mediated pathways using the drug cocktail also revealed a somewhat unexpected spread of lateral excitation. Under inhibitory receptor blockade, the light-evoked iGluSnFR responses continued to grow even when the spots sizes were increased significantly beyond the size of BC dendritic fields (~50 µm; *Figure 4A*, n=71 proximal and 431 distal ROIs, 5 FOVs, 4 retinas from 4 mice; *p<0.01). The maximal peak response was evoked for spots that were ~200 µm diameter. Interestingly, the amplitude of the plateau phase was further increased by ~23% in the proximal and ~20% in the distal dendritic sites, between 200 µm and 800 µm (*p<0.01, t-test). Such lateral excitation is likely to be attributed to electrical coupling, which occurs between BCs as well as amacrine cells (*Sigulinsky et al., 2020*; *Arai et al., 2010*; *Asari and Meister, 2014*). The finding that the kinetic diversity of BC responses is maintained in the absence of inhibition, suggests that they are shaped in large part by excitatory network mechanisms.

## BC output kinetics contribute to direction selectivity

Next, we tested how the apparent kinetic diversity in BC input impacts direction selectivity. Since the BC kinetics inferred from iGluSnFR measurements are in part dictated by the properties of the indicator, it remains unclear how they relate to the starbursts' physiological responses mediated by endogenous AMPA receptors. To address this issue, we first used optical deconvolution methods (*Awatramani et al., 2007*) to estimate the time-varying vesicle release rates from individual BCs and then used a computational model to understand how trains of vesicles are transformed into AMPA receptor-driven voltage signals in starburst dendrites.

Time-varying vesicle release rates from proximal and distal BCs were estimated by deconvolving the iGluSnFR response with an idealized quantal response described by an alpha function with decay kinetics ~30 ms (*Figure 5B*, inset), which was obtained by matching the kinetics of 'spontaneous' quantal responses (*Figure 5A*). The resulting release rates were discretized using a Poisson process, yielding an estimate of the temporal pattern of single vesicle release events (*Figure 5C*). For this analysis, the quantal size for each ROI was estimated using fluctuation analysis (see Materials and methods). Indeed, convolving the average quantal response with the inferred trains of vesicle release events (*Figure 5C*) produced a signal similar to the original iGluSnFR measurement (*Figure 5D*). The release profiles of proximal and distal BCs obtained in this manner support the notion that they have different capacities to release vesicles in response to light stimuli: proximal BCs release vesicles throughout the duration of the stimulus, while distal BCs release a large fraction of their vesicles at the onset of the light stimulus (*Figure 5E*).

Previous experimental and modeling studies have shown starburst dendritic sectors to be relatively electrically isolated from each other (*Poleg-Polsky et al., 2018*; *Morrie and Feller, 2018*; *Ozaita et al., 2004*). Since the potential for intra-dendritic signaling is likely to be minimal, we constructed a simple ball-and-stick model (NEURON), rather than a more complex network model. Using previously described properties of starburst dendrites (*Tukker et al., 2004*; *Vlasits et al., 2016*), we directly tested how vesicle release dynamics impacts starburst dendritic computations, which is hard to achieve experimentally.

Sustained and transient BCs were assumed to reflect the properties of BC7 and BC5s, respectively, and their positions on the starburst dendrites were picked pseudo-randomly from the distributions described previously by *Ding et al., 2016*; *Figure 6—source data 1*. The results shown are

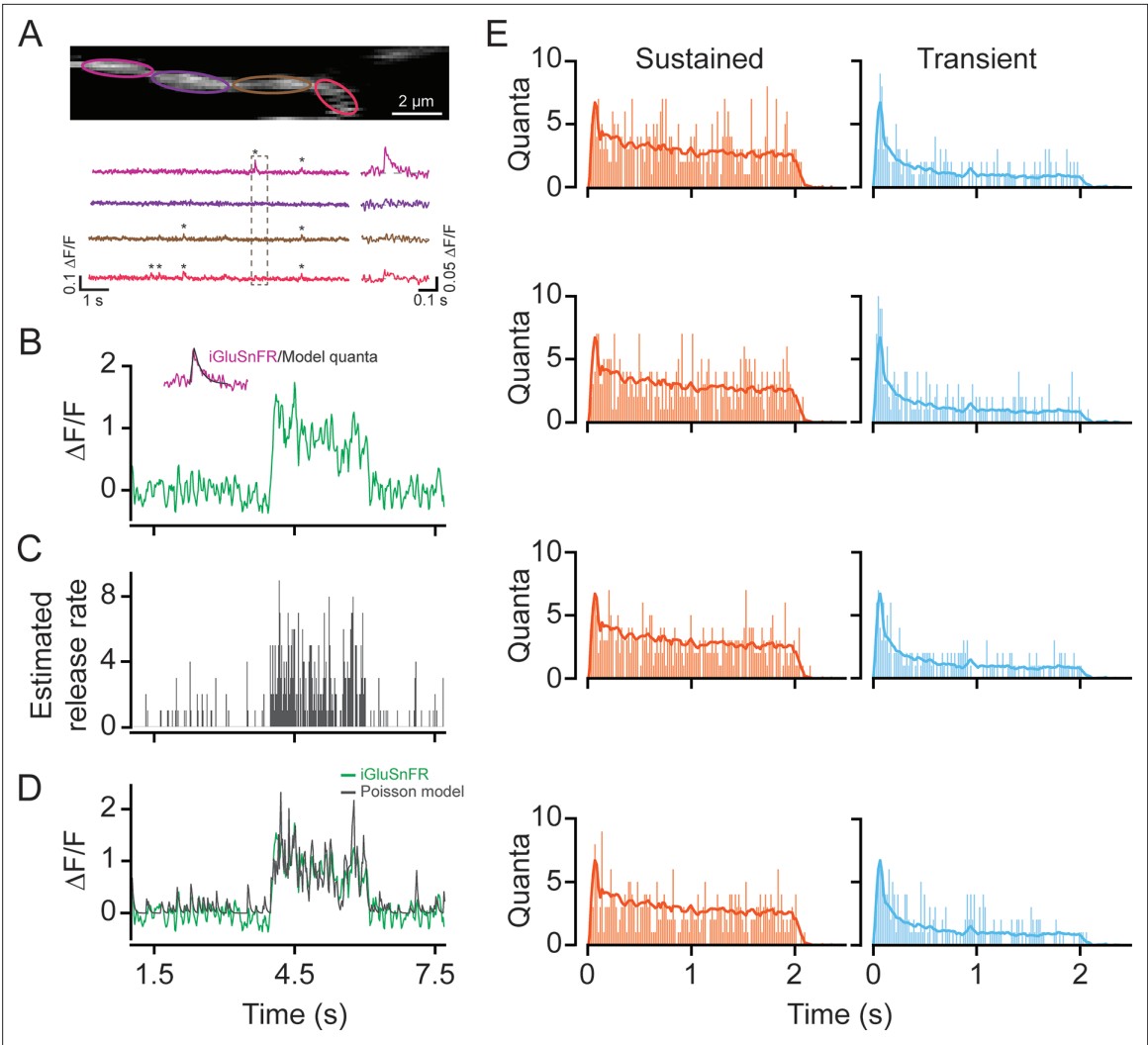

**Figure 5.** Time-varying vesicle release rates estimated using temporal deconvolution. (**A**) Spontaneous iGluSnFR signals measured in neighboring ROIs across a small dendritic section (color-coded to match ROIs). (**B**) A typical light-evoked iGluSnFR signal measured from proximal dendrites; inset: iGluSnFR quantal event fitted with an alpha function. (**C**) A time-varying release rate was estimated by deconvolving the iGluSnFR signal with the quantal signal (shown in (**B**)). (**D**) Convolving the estimated release rate with a unitary event recapitulates the shape of the original iGluSnFR response. (**E**) Example vesicle release rates for sustained (orange) and transient (blue) iGluSnFR responses. The solid line indicates the average vesicle release rates for all ROIs. n=50 each, proximal and distal ROIs. ROI, region of interest.

obtained from averaging many simulations, each with resampled BC complements (*Figure 6A*; also see Materials and methods). A simulated moving bar (400 μm wide) 'activated' BCs in succession, initiating streams of postsynaptic AMPA receptor-mediated miniature-like events ($I_{decay}$~0.54 ms; *Vlasits et al., 2016*) according to their location in the dendrite. In these simulations, we assumed the AMPA receptor-mediated events summed linearly (i.e., AMPA receptors did not saturate or desensitize during trains of activity). Synapses were sequentially turned off after the bar traversed BC receptive fields (60 μm diameter). Thus, the input duration at each point in the dendrite varied linearly with stimulus velocity.

The voltage responses measured from the model cell soma were qualitatively similar to those measured experimentally. For example, responses measured in the soma rose rapidly in the preferred direction compared to those evoked in the null direction, owing to the asymmetric distribution of inputs (*Figure 6A*, middle panel; *Ankri et al., 2020*). The model also accurately recapitulated direction selectivity measured in the distal intracellular $Ca^{2+}$ signals (*Figure 6A*, bottom traces) similar to results from two-photon $Ca^{2+}$ imaging experiments (*Euler et al., 2002*). As expected for a mechanism that relies on a fixed delay, this direction selectivity was strongly dependent on stimulus velocity. In our

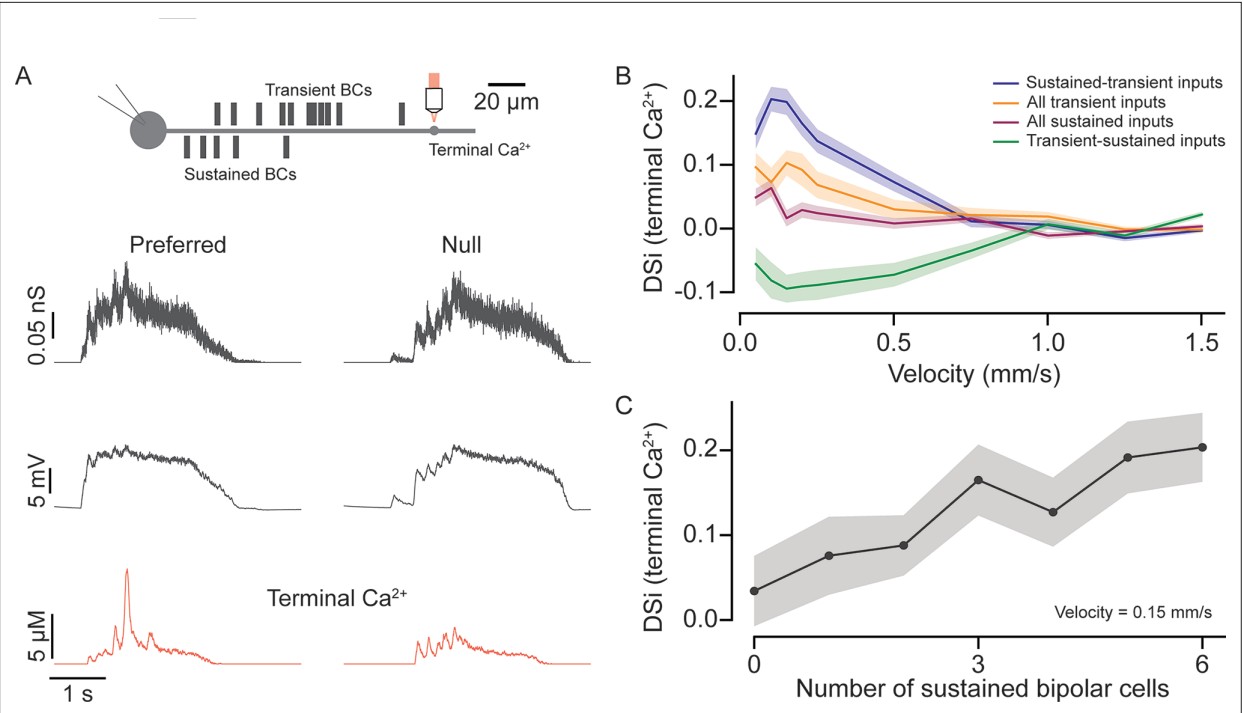

**Figure 6.** Input kinetics shape direction selectivity at low stimulus velocities. (**A**) Schematic representation of locations of somatic voltage and terminal Ca²⁺ recordings from a model SAC under simulated conditions (top). Bipolar cell conductances, somatic voltage, and terminal Ca²⁺ responses (bottom) were measured in the preferred and null direction from the model SAC when simulated using moving bars. (**B**) Direction selectivity index (DSi) of peak Ca²⁺ (terminal) responses versus velocity for different BC input distributions—(i) sustained and transient inputs; when proximal (sustained) and distal (transient) inputs are distributed based on connectomics data (original model); (ii) transient-sustained inputs; when sustained and transient BC inputs are reversed at their locations; (iii) all transient inputs; when all proximal inputs are replaced by transient BCs; and (iv) all sustained inputs; when all distal inputs are replaced by sustained BCs. Shading indicates ± s.e.m. (**C**) DSi of peak Ca²⁺ (terminal) responses versus number of inputs from sustained BCs at a velocity of 0.15 mm/s. BC, bipolar cell.

The online version of this article includes the following source data and figure supplement(s) for figure 6:

**Source data 1.** Model parameters.

**Figure supplement 1.** Relationship between the direction selectivity index and the magnitude of the spatial offset between bipolar cell (BC) inputs.

model based on the experimentally determined input kinetics, anisotropic summation occurs most robustly for slow-moving stimuli (<0.5 mm/s).

Four lines of evidence highlight the importance of BC kinetics in enhancing direction selectivity. First, when the kinetics of proximal and distal BCs were exchanged (i.e., proximal BCs were made transient and the distal BCs were made sustained), the directional preference was reversed from preferring centrifugal to centripetal motion (resulting in a negative DSi; *Figure 6B*). Notably, the strength of the directional preference was stronger for the native compared to the switched BC distributions, indicating that the kinetic mechanisms work in concert with intrinsic dendritic mechanisms (*Hausselt et al., 2007*; *Tukker et al., 2004*). Second, DSi was strongly decreased when all BCs were made either sustained or transient (*Figure 6B*). In all cases, the direction selectivity was most robust for slow-moving stimuli, and we found that these BC distribution manipulations had a statistically significant effect on DSi up to 1 mm/s ($p < 0.05$; one-way analysis of variance, ANOVA; *Figure 6B*). DSi was significantly higher in the condition when BC distribution followed the native sustained-transient distribution over when all BCs were sustained ($p < 0.0005$; t-test), for velocities up to 0.5 mm/s. Third, when we fixed the BC input locations and incrementally converted sustained input into transient ones (starting from the proximal site furthest from the soma), direction selectivity decreased linearly (*Figure 6C*; velocity 0.15 mm/s). This is presumably because the amount of asymmetric temporal summation decreases as inputs with weaker plateau phases are swapped in.

Finally, as the BC populations for each repeat were randomly distributed, we asked how much the average distance between the proximal and distal populations on a particular trial impacts the

resulting DSi (*Figure 6—figure supplement 1*). We found that for velocities with the strongest DS response (positive for the control condition, and negative for the swapped release profile condition), there were indeed linear relationships between the magnitude of DSi and the mean distance between the sustained and transient BC input ($R^2 \approx 0.208$ and 0.269, respectively, *Figure 6—figure supplement 1*). Thus, we conclude that in the context of realistic estimates of glutamate input kinetics, the 'space-time' wiring model for direction selectivity holds merit under specific stimulus conditions.

## Discussion

Recent models of direction selectivity in the primary sensing neurons in the fly visual system and mammalian retina incorporate key aspects of two classic models of direction selectivity reviewed by *Borst et al., 2020*, *Murphy-Baum et al., 2021*: one relying on multiplicative interactions between excitatory inputs (*Hassenstein and Reichardt, 1956*) and the other relying on divisive inhibition (*Barlow and Levick, 1965*). While the neural substrates underlying inhibition appear to be well worked out, defining the precise sources of excitation and their kinetics has proven to be more difficult. In this study, we used iGluSnFR imaging to directly estimate glutamate vesicle release rates along the length of starburst dendrites. This method circumvents many of the uncertainties associated with $Ca^{2+}$ or electrophysiological measurements from the soma that has typically been applied to assess the properties of input kinetics. Our results demonstrate stark kinetic differences in glutamate input across starburst dendrites, supporting the idea that nonlinear excitatory interactions enhance direction selectivity (*Kim et al., 2014*; *Greene et al., 2016*).

At first pass, the input kinetic diversity across starburst dendrites we observed appears to contradict recent large-scale surveys which show BC types of the same response polarity to have similar kinetics (*Franke et al., 2017*). However, this discrepancy is easily explained by the choice of precise stimulus parameters. Specifically, robust kinetic differences between responses were measured in proximal and distal regions across a range of spot sizes, except for the smallest size that just covered the BC receptive field, which was used by *Franke et al., 2017*. Indeed, when larger spots or full-field stimuli are utilized BC kinetics are found to vary across the inner-plexiform layer (*Borghuis et al., 2013*; *Gaynes et al., 2022*; *Franke et al., 2017*).

The precise mechanisms that give rise to the kinetic diversity in BC input remain to be identified. Previous studies have shown the functional diversification of BC responses arises mainly from GABAergic and/or glycinergic inhibition (*Franke et al., 2017*; *Franke and Baden, 2017*). Thus, it was surprising to observe clear kinetic differences when all inhibitory pathways were blocked. This implied that BC kinetics could be influenced by inhibition, but inhibition was not required to generate kinetic differences. Additionally, kinetic differences persist when rod-mediated signals are blocked by CNQX suggesting that kinetic differences cannot be attributed solely to changes associated with rod/cone vision, consistent with a previous report (*Awatramani and Slaughter, 2000*). The finding that input kinetic differences manifest for spot sizes larger than BC dendritic fields implied that network mechanisms (*Figure 4A*), rather than cell-intrinsic factors are involved, although these mechanisms are not mutually exclusive (*Awatramani and Slaughter, 2000*). Specific types of BCs are known to be coupled to each other as well as to selective amacrine cells including the AII and A8 amacrine cells (*Arai et al., 2010*; *Asari and Meister, 2014*; *Demb and Singer, 2012*; *Sigulinsky et al., 2020*; *Yadav et al., 2019*; *Zhang and Wu, 2009*). Such coupling has been implicated in extending BC receptive fields spatially in salamander and fish retinas (*Hare and Owen, 1990*; *Saito and Kujiraoka, 1988*), and in mice, the electrical coupling can non-linearly affect the output of individual BCs (*Kuo et al., 2016*). However, whether BC7 have stronger coupling patterns compared to BC5 remains to be tested.

Another major advance made in this study is to relate iGluSnFR signals to endogenous vesicle release rates from BC terminals and eventually to starburst function. This is not a trivial problem as the properties of iGluSnFR signals may depend on a variety of factors including the expression pattern, affinity, and kinetic properties of iGluSnFR, and in general expected to be vastly different from glutamate receptors that mediate the starburst synaptic responses. Previous studies have shown iGluSnFR signals reflect local glutamate release from individual BC terminals despite being expressed uniformly along starburst dendrites (*Borghuis et al., 2013*; *Franke et al., 2017*; *Gaynes et al., 2022*; *Matsumoto et al., 2021*; *Strauss et al., 2022*). Here, we were also able to measure spontaneous 'quantal-like' events using high-speed imaging, which enabled estimates of vesicle release rates using temporal deconvolution. This analysis assumes that the light-evoked responses are made up of a

linear superposition of quantal events and there is not a significant build-up of glutamate within the synaptic cleft during ongoing activity. With these assumptions in mind, we estimated the peak instantaneous release rate to be 5–10 vesicles/s, which is in line with previous in vivo estimates made in zebrafish BC terminals (*James et al., 2021*; *James et al., 2019*). We also estimated the steady-state release rates to be ~3 vesicles/s at proximal sites and ~1 vesicle/s at distal sites, which is well below the maximum release rate estimated for photoreceptor synapses, which also contain ribbons (*Hays et al., 2021*; *Hays et al., 2020*).

Estimating the vesicle release rates at single dendritic sites was an important step because it allowed us to build simple a biophysical model to probe how starburst dendrites respond to distinct patterns of vesicle release evoked by stimuli moving in different directions, using their endogenous AMPA receptors. In a circuit model that is constrained by experimentally determined vesicle release rates and by the spatial distributions of sustained and transient input to starburst dendrites, we found that differences in input kinetics mainly enhance direction selectivity at low velocities (<500 μm/s). Our modeling approach contrasts with recent studies in which key parameters including the vesicle release rates and the specific distribution of BC types along starburst dendrites were determined by an algorithm that optimizes direction selectivity (*Ezra-Tsur et al., 2021*). It also contrasts with phenomenological models such as the linear-non-linear cascade models, which leave out the biophysical details of the starburst (*Kim et al., 2014*; *Franke et al., 2017*; *Strauss et al., 2022*; *Fransen and Borghuis, 2017*). It should be noted, however, that the intrinsic properties of the starburst remain poorly defined and future work is needed to experimentally verify how inputs with distinct kinetics are transformed into dendritic voltage responses. In addition, the robust direction selectivity that is experimentally observed in starbursts occurs across the range of velocities (100–2000 μm/s; *Ding et al., 2016*; *Summers and Feller, 2022*), indicating additional mechanisms are involved. Further work is needed to establish how the excitatory and inhibitory network mechanisms work together to produce direction selectivity that is invariant across a large range of velocities. It is also worth noting while our study emphasizes the role of BC kinetics in driving direction selectivity for slow-moving spots, for other stimuli, such as radial motion, or for stimuli that emerge suddenly, the inhibitory surrounds of BCs may enhance motion sensitivity in more complicated ways (*Gaynes et al., 2022*; *Strauss et al., 2022*).

Space-time wiring models for direction selectivity have been proposed in diverse systems including the primate retina (*Patterson et al., 2022*; *Kim et al., 2022*; *Smith et al., 2022*), visual cortex (*Lien and Scanziani, 2018*; *De Valois and Cottaris, 1998*), as well as the fly optic lobe (*Haag et al., 2016*; *Leong et al., 2016*; *Behnia et al., 2014*; *Takemura et al., 2013*; but see *Takemura et al., 2017*). Recent studies have made great strides in identifying the precise anatomical substrates underlying the computation, in all these model systems. Elegant studies assessing presynaptic dynamics, provide strong support that anisotropic summation of excitatory signals shapes direction selectivity (*Lien and Scanziani, 2018*; *Groschner et al., 2022*). Future studies using optical sensors will directly confirm whether the observed patterns of presynaptic activity are transformed into the distinct temporal patterns of excitation required to generate direction selectivity at the level of dendrites of DS neurons, as we have observed here for starburst dendrites in the mouse retina.

## Materials and methods
### Animals
Experiments were performed using ChAT-IRES-Cre (Δneo) mice on a C57BL/6J background (Jackson Laboratory, #031661), in which the IRES-Cre was targeted to the *ChAT* locus. Mice were P21 or older and of either sex. Animals were housed under 12-hr light/dark cycles. All procedures were performed in accordance with the Canadian Council on Animal Care and approved by the University of Victoria's Animal Care Committee, or in accordance with Danish standard ethical guidelines and were approved by the Danish National Animal Experiment Committee (Permission nos. 2015-15-0201-00541 and 2020-15-0201-00452).

### Viral injections
For intravitreal injections, mice were anesthetized by administering isoflurane (2–3% at 1–1.5 L/min; Fresenius Kabi Canada Ltd) mixed with oxygen (1–3%) through a vaporizer. Buprenorphine was

administered subcutaneously (0.05–0.1 mg/kg body weight) as an analgesic. After creating a small hole at the margin of the sclera and cornea with a 30-gauge needle, a volume of 1–1.2 μl of the viral plasmid pAAV.hSyn.Flex.iGluSnFR.WPRE.SV40 (gift from Dr. Loren Looger; Addgene plasmid #98931; http://n2t.net/addgene:98931; RRID:Addgene_98931) was injected into the vitreous humor of either the left or right eye using a Hamilton syringe (syringe: 7633-01, needle: 7803-05, point style 3, and length 10 mm). Mice were returned to their home cage after their complete recovery, which was facilitated using a heating pad. Imaging experiments were performed at least 3 weeks after injections.

## Tissue preparation

Mice were dark-adapted for at least 60 min before being anesthetized with isoflurane (Fresenius Kabi Canada Ltd) and decapitated. Retinas were extracted in Ringer's solution under a dissecting microscope equipped with infrared optics, and flat-mounted onto a glass poly-L-lysine (Sigma-Aldrich) coated coverslip before being placed into the recording chamber, perfused with oxygenated Ringer's solution (95% $O_2$/5% $CO_2$; 35°C). Retinas were visualized with a Spot RT3 CCD camera (Diagnostics Instruments) through a 40× or 60× water-immersion objective on a BX-51 WI microscope (Olympus Canada).

## Two-photon imaging

For iGluSnFR imaging, we used an Insight DeepSee+ laser (Spectra-Physics) tuned to 920 nm, which was guided by an 8 kHz resonant-galvo-galvo mirror set (Vidrio Technology). Green and red fluorescent signals were detected using photomultiplier tubes (PMTs; Hamamatsu) that were equipped with appropriate bandpass filters (525/45 and 625/90, respectively; Semrock). Single-photon events were acquired using a high-speed current amplifier (200 MHz, Edmund Optics), and converted into images using ScanImage software (*Pologruto et al., 2003*). While most recordings from proximal and distal dendritic regions were acquired consecutively at a frame rate of 58.25 Hz (256×256 pixels), in some experiments the responses from the two regions were recorded near-simultaneously at a frame rate of 22.5 Hz (256×256 pixels) using an electrically tunable lens (ETLs; EL-10-30-TC-NIR-12D, Optotune; see Materials and methods; *Murphy-Baum and Awatramani, 2022*). Data using either method were compiled in *Figure 2C–D*.

For iGluSnFR imaging from T7 BCs, the isolated retina was placed under the microscope (Slice-Scope, Scientifica) equipped with a galvo-galvo scanning mirror system, a mode-locked Ti: Sapphire laser tuned to 940 nm (MaiTai DeepSee, Spectra-Physics). The iGluSnFR signals emitted were passed through a set of optical filters (ET525/50m, Chroma; lp GG495, Schott) and collected with a GaAsP detector. Images were acquired at 8–10 Hz using custom software developed by Zoltan Raics (SENS Software).

## Visual stimulation

Visual stimuli were produced using a digital light projector and were focused onto the photoreceptor layer of the retina through the sub-stage condenser. Light stimuli were presented during the brief turn-around phase of the resonant mirror and thus did not interfere with the collection of fluorescent signals. The background illuminance was measured to be ~1000 photon/μm$^2$/s. Visual stimuli were generated and presented using StimGen, a python-based visual stimulation interface (https://github.com/benmurphybaum/StimGen, archived at https://doi.org/10.5281/zenodo.7331820; *Murphy-Baum, 2022*). For all experiments, static spots of different sizes (100, 200, 400, and 800 μm diameter) were presented for a duration of 2 s. Stimuli were presented 4 s after the start of two-photon image acquisition.

## Pharmacology

The following concentrations (in μM) of antagonists were used for experiments: 20 CNQX disodium salt (Hello Bio, UK), 100 TPMPA (Tocris Bioscience), and 5 SR-95531 (Hello Bio, UK). Antagonists were initially prepared as stock solutions in distilled water or DMSO in the case of CNQX. During experiments, drugs were freshly prepared from a stock solution in carboxygenated Ringer's solution. All drug solutions were bath applied to the tissue for at least 5 min prior to recording.

## Quantification and statistical analysis

All imaging analysis and statistical comparisons were performed in Igor Pro (WaveMetrics). Fluorescence signals were measured in small ROIs (5×5 μm$^2$ or 10×10 μm$^2$) placed along single dendrites, or

in a grid-like pattern over the starburst dendritic plexus. The size of these ROIs was chosen to ensure adequate signal-to-noise ratio (SNR) computed as:

$$SNR = \frac{Peak\ value}{s.d.\ of\ the\ baseline} \tag{1}$$

where the standard deviation (s.d.) of the baseline fluorescence was measured in a 1-s window before the start of the stimulus. Only ROIs with SNR>4 were selected for further analysis.

The sustained transient index (STi) for each ROI was calculated as:

$$STi = \frac{\frac{\Delta F}{F}\ Plateau\ phase}{\frac{\Delta F}{F}\ Peak} \tag{2}$$

where the ΔF/F for the plateau phase was averaged over the last 1 s of the light response.

## Release rate estimates

Discrete Fourier transforms of responses (*Response*) to stationary spots of light were divided by the Fourier transforms of the idealized quantal response (*Quantum*). The inverse Fourier transform of the resulting ratio yielded estimates of the instantaneous release rate (RR) in the time domain from representative BCs, as previously described (*Van der Kloot, 1988*; *Diamond and Jahr, 1995*):

$$RR = IFFT\left(\frac{FFT(Response)}{FFT(Quantum)}\right) \tag{3}$$

The temporal waveform for the idealized Quantum was obtained by fitting averaged spontaneous iGluSnR events with exponential functions (2 ms rise time and 30 ms decay time constants). The amplitude of the Quantum for each ROI was set according to the quantal size estimate (QSE). The QSE is defined following classical quantal analysis (*Katz and Miledi, 1972*), as:

$$QSE = \frac{2\sigma^2}{\mu} \tag{4}$$

where $\sigma$ and $\mu$ are the variance and the mean of the steady state of iGluSnFR responses, respectively. Averages responses from proximal and distal scan fields were used to calculate the prototypical sustained and transient release rates, which were assumed to correspond to BC7 and BC5s, respectively (n=50 ROIs, each proximal and distal site, 6 retinas, 7 FOVs).

## Computational modeling

A simple ball and stick compartmental model was constructed in the NEURON simulation environment (https://github.com/geoffder/spatiotemporal-starburst-model; copy archived at *de Rosenroll, 2022*), with one somatic compartment and three dendritic compartments (initial, middle, and terminal dendrites; *Ding et al., 2016*) with membrane properties and channels similar to those published previously (*Ding et al., 2016*; *Vlasits et al., 2016*; see *Figure 6—source data 1*). Input to this isolated starburst dendritic computational unit was driven by model BC synapses that were pseudo-randomly distributed based on anatomical measurements (*Ding et al., 2016*). Specifically, for each trial repetition, 6 proximal locations and 12 distal locations were sampled from the probability density functions for BC7 and BC5s synapse locations, respectively (*Ding et al., 2016*) (see *Figure 6—source data 1*). To simulate moving light bars, BC inputs were activated in sequence based on their location on the dendrite. Vesicle release from each BC ceased when the bar left its receptive field (60 μm in diameter). The precise temporal sequence of BC activation and the response duration was set to simulate 400 μm wide light bars moving over a range of velocities (0.1–2 mm/s).

For each synapse, on each stimulus presentation, the timing of each vesicle release event from individual BCs was obtained by discretizing the continuous release rates estimated from the deconvolution analysis via a Poisson process (sampled at a model time step of 1 ms). Glutamate released from each vesicle then activated AMPA receptor-mediated 'miniature' events modeled using a double exponential function (rise 0.14 ms, decay 0.54 ms, and reversal 0 mV; *Vlasits et al., 2016*). The synaptic conductances were scaled down linearly from 172.2 pS to 68.6 pS across the dendrite, to mimic the downward trend of experimentally measured glutamate responses (*Vlasits et al., 2016*). Finally, changes in the somatic membrane voltage and terminal dendritic $Ca^{2+}$ accumulation in response to moving bars were recorded to assess post-synaptic activation (*Jain et al., 2022*; *Ding et al., 2016*;

*Tukker et al., 2004*). The direction-selective index (DSi) of the model was assessed by subtraction of the peak terminal Ca²⁺ concentration in centripetal stimulation from that of centrifugal stimulation.

To quantify the temporal kernels extracted using reverse correlation (*Figure 3—figure supplement 2*), we calculated the biphasic peak and area index as:

$$Trough = max\left(0, min\left(0.5 - min\left(contrast\right), 1\right)\right)$$

$$Peak = max\left(0, min\left(max\left(contrast\right) - 0.5, 1\right)\right)$$

$$Biphasic\ peak\ index = \frac{Trough}{(Peak+Trough)}$$

$$negative = \sum\left(max\left(min\left(-1, contrast - 0.5\right), 0\right)\right)$$

$$positive = \sum\left(max\left(min\left(0, contrast - 0.5\right), 1\right)\right)$$

$$Biphasic\ area\ index = \frac{-negative}{(positive-negative)}$$

## Acknowledgements

The authors thank Dr. Laura Hanson (UVic) for spontaneous iGluSnFR measurements, members of the Awatramani lab for their helpful discussions, and Zoltan Raics for software development. This work was supported by a CIHR grant (awarded to GBA).

---

## Additional information

### Funding

| Funder | Grant reference number | Author |
|---|---|---|
| Canadian Institutes of Health Research | 159444 | Gautam Bhagwan Awatramani |
| European Research Council | 638730 | Keisuke Yonehara |

The funders had no role in study design, data collection and interpretation, or the decision to submit the work for publication.

### Author contributions

Prerna Srivastava, Conceptualization, Data curation, Formal analysis, Investigation, Visualization, Methodology, Writing - original draft, Writing – review and editing; Geoff de Rosenroll, Software, Formal analysis, Writing – review and editing, Computational modeling; Akihiro Matsumoto, Data curation, Formal analysis; Tracy Michaels, Resources; Zachary Turple, Conceptualization; Varsha Jain, Gautam Bhagwan Awatramani, Conceptualization, Data curation, Supervision, Funding acquisition, Visualization, Methodology, Writing - original draft, Writing – review and editing; Santhosh Sethuramanujam, Keisuke Yonehara, Conceptualization, Supervision, Writing – review and editing; Benjamin L Murphy-Baum, Methodology, Writing – review and editing

### Author ORCIDs

Prerna Srivastava (iD) http://orcid.org/0000-0002-3429-7039
Geoff de Rosenroll (iD) http://orcid.org/0000-0002-5431-2814
Benjamin L Murphy-Baum (iD) http://orcid.org/0000-0001-6746-3091
Gautam Bhagwan Awatramani (iD) http://orcid.org/0000-0002-0610-5271

### Ethics

All procedures were performed in accordance with the Canadian Council on Animal Care and approved by the University of Victoria's Animal Care Committee, or in accordance with Danish standard ethical guidelines and were approved by the Danish National Animal Experiment Committee (Permission No. 2015-15-0201-00541; 2020-15-0201-00452).

### Decision letter and Author response

Decision letter https://doi.org/10.7554/eLife.81533.sa1

Author response https://doi.org/10.7554/eLife.81533.sa2

## Additional files

### Supplementary files
• MDAR checklist

### Data availability
Source data has been provided. The following GitHub repository contains modelling files: https://github.com/geoffder/spatiotemporal-starburst-model (copy archived at swh:1:rev:7ee665a982e1fe3b075df3eb2e6129afd5dc8055).

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
