## [Editor Report]

This is an important paper that addresses a key mechanism that underlies the canonical computation of direction selectivity in the retina. By using fluorescence imaging of glutamate release from excitatory interneurons combined with a computational model of dendritic integration, the authors make a convincing case that the kinetics of glutamate release contributes to the direction-selectivity of individual neural processes in retinal neurons. This work will appeal to visual neuroscientists as well as cellular physiologists interested in dendritic computations.

---

## [Decision Letter]

**Decision letter after peer review:**

Thank you for submitting your article "Spatiotemporal properties of glutamate input support direction selectivity in the dendrites of retinal starburst amacrine cells" for consideration by *eLife*. Your article has been reviewed by 3 peer reviewers, and the evaluation has been overseen by a Reviewing Editor and Joshua Gold as the Senior Editor. The following individual involved in the review of your submission has agreed to reveal their identity: Katrin Franke (Reviewer #3).

Essential revisions:

1) Increase the n for experiments based on individual SAC dendrites. Reviewers note that this experiment summarized in Figure 1 is underpowered.

2) There are several suggested revisions to the model – most of these suggestions can be addressed with a bit more explanation of the rationale, though there are some more substantive suggestions regarding the use of ball-and-stick models for dendrites. I am hoping that you read the reviewers' suggestions with an open mind as to what you think will best convey the primary point you'd like to make and explain these implemented changes in a way that addresses the issues raised by the reviewers. To be clear, you do not need to do everything all reviewers suggested regarding the model but please offer an explanation for your choices in response to reviewers' specific concerns.

*Reviewer #2 (Recommendations for the authors):*

The BC7 and deconvolution experiments in Figures 3 and 5 are very nice, but they come across as a bit descriptive/anecdotal in the absence of analyzed pooled data.

Given the nature of the conclusions, it would seem appropriate to recognize in the Discussion Hassenstein and Reichardt (1956) and at least one of Barlow's papers on the subject.

*Reviewer #3 (Recommendations for the authors):*

– In Suppl. Figure 2, the authors show that temporal response properties of type 7 bipolar cells do not change with signal to noise of the responses. They should also show the same analysis for the population data, together with the distribution of signal-to-noise ratios of proximal and distal ROIs. This will make sure that differences observed for proximal versus distal ROIs are unrelated to differences in response strength.

– The authors should provide more details in the Methods section regarding ROI detection, the rationale behind ROI sizes, and the fraction of ROIs that passed their threshold. In addition, I suggest not using the peak value, but instead a percentile for estimating the signal-to-noise ratio, as this is more robust against outliers.

– In Figure 6C, the authors should indicate which velocities are significantly different between the two conditions.

– The first sentence of the abstract reads like the space-time-wiring model is the main proposed mechanism underlying direction selectivity. I suggest the authors rephrase this sentence to illustrate that this is one of many, well-described other mechanisms.

– In the Discussion section, the authors write: "Moreover, kinetic differences were discernable for stationary or slow-moving spots." I suggest rephrasing, as the slow-moving spot result was only predicted by their model.

---

## [Author Response]

Reviewer #2 (Recommendations for the authors):The BC7 and deconvolution experiments in Figures 3 and 5 are very nice, but they come across as a bit descriptive/anecdotal in the absence of analyzed pooled data.

We estimated the time-varying release rates by deconvolving the iGluSnFR signals (obtained by the experimental data) with the quantal signal. The release rates are pooled from the iGluSnFR experimental data and the comparisons with the Poisson trains are simply to represent that the estimated release rates can be used to generate the shape of the original iGluSnFR responses. To estimate the kinetics for our model BCs, we pooled data from 50 ROIs for proximal and distal sites (taken from, 6 retinas, 7 FOVs).

BC7 experiments were performed to bolster the idea that type 7 BCs are the primary source of sustained responses in the proximal dendritic region of SACs, which more effectively supports the ‘space-time’ wiring model.

Given the nature of the conclusions, it would seem appropriate to recognize in the Discussion Hassenstein and Reichardt (1956) and at least one of Barlow's papers on the subject.

Agreed! Thanks.

Reviewer #3 (Recommendations for the authors):– In Suppl. Figure 2, the authors show that temporal response properties of type 7 bipolar cells do not change with signal to noise of the responses. They should also show the same analysis for the population data, together with the distribution of signal-to-noise ratios of proximal and distal ROIs. This will make sure that differences observed for proximal versus distal ROIs are unrelated to differences in response strength.

Agreed. The Sustained/transient index for proximal and distal ROIs is plotted against signal amplitude in Figure 3 —figure supplement 1. This plot shows that the kinetic properties are not confounded by signal-to-noise issues.

– The authors should provide more details in the Methods section regarding ROI detection, the rationale behind ROI sizes, and the fraction of ROIs that passed their threshold. In addition, I suggest not using the peak value, but instead a percentile for estimating the signal-to-noise ratio, as this is more robust against outliers.

In recent studies examining BC properties the spatial extent of ROIs has ranged from 5 µm (Franke et al., 2017) to ~50 µm (Gaynes et al., 2022). In most experiments we found using 10 µm x 10 µm or 5 x 5 µm ensured a large fraction of ROIs had SNR > 4. We found a weak relationship between response amplitude and kinetics (data now shown in Figure 3 —figure supplement 1). Since our estimates of iGluSnFR response kinetics did not appear to be seriously confounded by noise, we did not use the alternate method suggested by the Reviewer. We have described our methods more clearly in the text.

– In Figure 6C, the authors should indicate which velocities are significantly different between the two conditions.

We used a one-way ANOVA to demonstrate that there exists a group effect across all velocities i.e., changing the bipolar release profiles has a statistically significant effect. In the model data, subsequent pairwise comparisons at individual velocities are less informative and we simply state that the effects of BC kinetics dwindle at higher speeds (0.5 mm/s). For example, a statistically significant DS can be measured at all but the highest velocities for control conditions vs single BC input conditions, but above 500um/s this DS is tiny and less meaningful.

– The first sentence of the abstract reads like the space-time-wiring model is the main proposed mechanism underlying direction selectivity. I suggest the authors rephrase this sentence to illustrate that this is one of many, well-described other mechanisms.

We have now changed the text accordingly.

– In the Discussion section, the authors write: "Moreover, kinetic differences were discernable for stationary or slow-moving spots." I suggest rephrasing, as the slow-moving spot result was only predicted by their model.

We have changed the text as per the Reviewer’s suggestion.